# Novel insights into breast cancer copy number genetic heterogeneity revealed by single-cell genome sequencing

Timour Baslan[1,2†]*, Jude Kendall[1], Konstantin Volyanskyy[3], Katherine McNamara[4], Hilary Cox[1], Sean D'Italia[1], Frank Ambrosio[1], Michael Riggs[1], Linda Rodgers[1], Anthony Leotta[1], Junyan Song[1,5], Yong Mao[3], Jie Wu[3], Ronak Shah[6], Rodrigo Gularte-Mérida[7], Kalyani Chadalavada[8], Gouri Nanjangud[8], Vinay Varadan[9], Assaf Gordon[10], Christina Curtis[4], Alex Krasnitz[1], Nevenka Dimitrova[3], Lyndsay Harris[9,11,12‡], Michael Wigler[1], James Hicks[1§]*

[1]Cold Spring Harbor Laboratory, Cold Spring Harbor, United States; [2]Department of Molecular and Cellular Biology, Stony Brook University, Stony Brook, United States; [3]Philips Research North America, Biomedical Informatics, Cambridge, United States; [4]Department of Genetics, Stanford University School of Medicine, Stanford, United States; [5]Department of Applied Mathematics and Statistics, Stony Brook University, Stony Brook, United States; [6]Center for Molecular Oncology, Memorial Sloan Kettering Cancer Center, New York, United States; [7]Department of Surgery, Memorial Sloan Kettering Cancer Center, New York, United States; [8]Molecular Cytogenetics Core Facility, Memorial Sloan Kettering Cancer Center, New York, United States; [9]Case Comprehensive Cancer Center, Case Western Reserve University, Cleveland, United States; [10]House Gordon Software Company LTD, Calgary, Canada; [11]Division of Hematology/Oncology, Department of Medicine, Case Western Reserve University School of Medicine, Cleveland, United States; [12]Seidman Cancer Center, University Hospitals of Case Western, Cleveland, United States

*For correspondence:
baslant@mskcc.org (TB);
jameshic@usc.edu (JH)

Present address: †Cancer Biology and Genetics Program, Memorial Sloan Kettering Cancer Center, New York, United States; ‡National Institutes of Health, Bethesda, United States; §USC Dana and David Dornsife College of Letters, Arts, and Sciences, University of Southern California, Los Angeles, United States

**Abstract** Copy number alterations (CNAs) play an important role in molding the genomes of breast cancers and have been shown to be clinically useful for prognostic and therapeutic purposes. However, our knowledge of intra-tumoral genetic heterogeneity of this important class of somatic alterations is limited. Here, using single-cell sequencing, we comprehensively map out the facets of copy number alteration heterogeneity in a cohort of breast cancer tumors. Ou/var/www/html/elife/12-05-2020/backup/r analyses reveal: genetic heterogeneity of non-tumor cells (i.e. stroma) within the tumor mass; the extent to which copy number heterogeneity impacts breast cancer genomes and the importance of both the genomic location and dosage of sub-clonal events; the pervasive nature of genetic heterogeneity of chromosomal amplifications; and the association of copy number heterogeneity with clinical and biological parameters such as polyploidy and estrogen receptor negative status. Our data highlight the power of single-cell genomics in dissecting, in its many forms, intra-tumoral genetic heterogeneity of CNAs, the magnitude with which CNA heterogeneity affects the genomes of breast cancers, and the potential importance of CNA heterogeneity in phenomena such as therapeutic resistance and disease relapse.

**eLife digest** Cells in the body remain healthy by tightly preventing and repairing random changes, or mutations, in their genetic material. In cancer cells, however, these mechanisms can break down. When these cells grow and multiply, they can then go on to accumulate many mutations. As a result, cancer cells in the same tumor can each contain a unique combination of genetic changes.

This genetic heterogeneity has the potential to affect how cancer responds to treatment, and is increasingly becoming appreciated clinically. For example, if a drug only works against cancer cells carrying a specific mutation, any cells lacking this genetic change will keep growing and cause a relapse. However, it is still difficult to quantify and understand genetic heterogeneity in cancer.

Copy number alterations (or CNAs) are a class of mutation where large and small sections of genetic material are gained or lost. This can result in cells that have an abnormal number of copies of the genes in these sections. Here, Baslan et al. set out to explore how CNAs might vary between individual cancer cells within the same tumor.

To do so, thousands of individual cancer cells were isolated from human breast tumors, and a technique called single-cell genome sequencing used to screen the genetic information of each of them. These experiments confirmed that CNAs did differ – sometimes dramatically – between patients and among cells taken from the same tumor. For example, many of the cells carried extra copies of well-known cancer genes important for treatment, but the exact number of copies varied between cells. This heterogeneity existed for individual genes as well as larger stretches of DNA: this was the case, for instance, for an entire section of chromosome 8, a region often affected in breast and other tumors.

The work by Baslan et al. captures the sheer extent of genetic heterogeneity in cancer and in doing so, highlights the power of single-cell genome sequencing. In the future, a finer understanding of the genetic changes present at the level of an individual cancer cell may help clinicians to manage the disease more effectively.

## Introduction

Research into the genetics of breast tumors has yielded comprehensive portraits of the somatic alterations acquired during the evolution of breast cancer genomes. Catalogues of recurrent driver alterations have been identified using an array of technologies (*Teixeira et al., 2002*; *Adeyinka et al., 2003*; *Chin et al., 2006*; *Sjöblom et al., 2006*; *Fridlyand et al., 2006*; *Banerji et al., 2012*; *Cancer Genome Atlas Network, 2012*; *Shah et al., 2012*; *Ciriello et al., 2015*; *Nik-Zainal et al., 2016*). This information has been instrumental in furthering our understanding of the basic biology that underlies breast cancer development and has led to the development of diagnostic and prognostic tests and more importantly, the development of efficacious targeted therapies (*Dawson et al., 2013*). While the adoption of therapeutic strategies targeting somatic cancer alterations and the pathways they control has had a profound impact on the treatment of breast cancer (*Alvarez et al., 2010*), disease relapse and therapeutic resistance remain important challenges (*Ma et al., 2015*; *Majewski et al., 2015*). Recent research into these phenomena has implicated genetic heterogeneity (i.e. sub-clonal variation or intra-tumoral heterogeneity) as a common mechanism to explain recurrence and treatment failure (*Toy et al., 2013*; *Carey et al., 2016*; *Miller et al., 2016*). However our knowledge of intra-tumoral genetic heterogeneity is limited and thus advancing it is instrumental in combating cancer.

CNAs are an important class of somatic mutations that are acquired during the evolution of breast cancer genomes (*Dawson et al., 2013*). Studies exploring the landscape of CNAs have considerably advanced our knowledge of breast cancer biology, with translational efforts leading to advances in the clinic. Most notably, the amplification of the ERBB2 (HER2) oncogene defines a biological subtype of breast cancers and targeting this CNA has led to the development of a multitude of efficacious therapeutic agents that have dramatically increased the survival of HER2+ patients (*Arteaga and Engelman, 2014*). Additionally, numerous studies have demonstrated the utility of copy number information in the prognostic stratification of breast tumors (*Russnes et al., 2010*; *Curtis et al., 2012*) and studies of the genes targeted by this class of somatic mutations have

substantially advanced our understanding of breast cancer biology (*Gatza et al., 2014*; *Cai et al., 2016*). However, our knowledge of CNA intra-tumoral genetic heterogeneity is limited and given the importance of this class of somatic alterations warrants further investigation.

Single-cell DNA sequencing methods have recently emerged as powerful tools for the study of intra-tumoral heterogeneity with recent applications in breast (*Navin et al., 2011*; *Eirew et al., 2015*) and other tumors (*Francis et al., 2014*; *Bolhaqueiro et al., 2019*; *Bian et al., 2018*) revealing novel cancer genetics and biology. Here, we present a comprehensive analysis of CNAs in breast cancer based on the sequencing of thousands of single-cell genomes across a cohort of breast tumors. Our results offer in-depth views of the magnitude of intra-tumoral CNA heterogeneity present in breast cancer genomes, reveal novel and important observations that have been missed by earlier bulk studies, highlight the unique nature of the data and its ability to retrieve novel knowledge, and provide an important foundation for the future exploration of intra-tumoral genetic heterogeneity of this important class of somatic mutations.

## Results

### Samples, annotations and sequencing

Samples from sixteen patients were selected from a cohort population enrolled in two phase II open-label clinical trials conducted by the Brown University Oncology Group (BrUOG). Two pre-treatment, freshly frozen biopsies per patient were obtained. Two blood samples from normal healthy individuals were also obtained and processed (discussed below – Methods). For the tumor samples, one biopsy was subject to bulk DNA and RNA isolation for transcriptome sequencing and copy number profiling by sparse sequencing. The second biopsy was subjected to single-cell copy number analysis as described previously (*Baslan et al., 2012*; *Baslan et al., 2015*). In brief, tissue was processed to obtain a suspension of DAPI stained single-nuclei. Nuclei were subsequently flow sorted by ploidy (measured by DNA-content) for single-nucleus deposition, genome amplification and sequencing. For samples with multi-modal ploidy distributions (i.e. diploid and polyploid populations) (*Figure 1—figure supplement 1A*), single-nuclei were sorted and sequenced from each distribution. A mean of 116 single-nuclei per tumor sample were sequenced in multiplex fashion for a total of 2086 single-cell genomes. For each nucleus, we targeted a sequencing depth of roughly 2 million reads, sufficient to call large copy number events (>1 MB) as well as focal events such as amplifications and deletions at a resolution of roughly 300kbs . Selected patient samples were representative of various biological and clinical variables such as PAM50 subtype, ploidy, and ER/PR/HER2 status as well as other parameters such as age and tumor size (*Figure 1—figure supplement 1B–D*).

### CNAs and genetic heterogeneity of stromal, non-cancer cells

Nuclei processed from the diploid (2N) distribution of sorted samples could either represent cancer cells or normal cells present in the tumor mass (i.e. normal epithelia, fibroblasts, and/or immune cells). When sequencing 2N nuclei of patient samples, we obtain copy number neutral, 'flat' genome profiles representative of non-cancer, stromal cells (*Figure 1—figure supplement 2A*). Intriguingly, we observed recurrent focal homozygous deletions on chromosomes 14 and 7, in otherwise diploid genomes (*Figure 1A*). Further inspection localized the deletions to T-cell receptor alpha (TCRa) and beta (TCRb) loci (*Figure 1—figure supplement 2B*). These deletions are the result of T-cell maturation and generated via the process of V-D-J recombination. Similarly, we observed focal homozygous deletions at the heavy and light Immunoglobulin loci (IGH and IGL) on chromosomes 14 and 22, respectively, denoting B-cells (*Figure 1B*). Thus, using single-nuclei copy number sequencing we can detect T- and B-cells present within the tumor mass. To gauge the sensitivity of T- and B-cell detection, we retrieved highly purified CD4+/CD8+ T-cells and CD27+/CD13+ B-cells and performed single-nuclei copy number analysis on approximately 96 nuclei for both samples with the expectation that all single-nuclei should carry their respective deletions. Deletion analysis led to a calculated detection sensitivity of ~80% and 50% for T- and B-cells, respectively (*Figure 1—figure supplement 2C*). This allowed the quantification of T- and B-cells in the tumor mass in our samples and led to the detection of T-cells (in varying proportions) in almost all patient samples, with B-cell

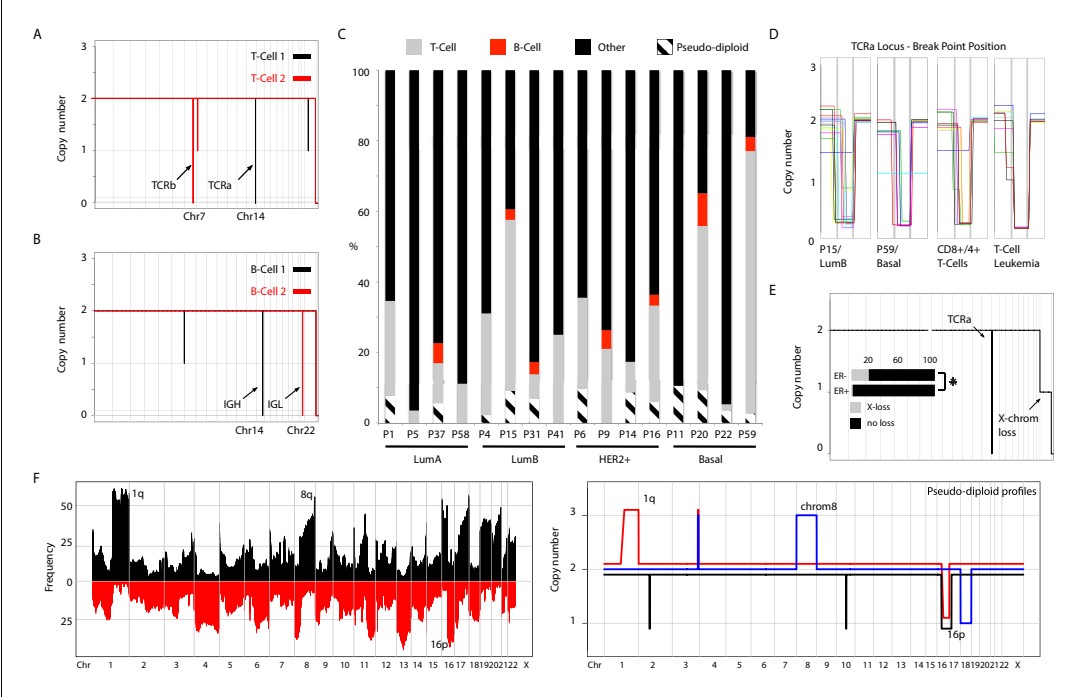

**Figure 1.** Genetic heterogeneity of immune cells and pseudo-diploid cells in breast cancer biopsies. (**A**) Genome-wide copy number view of representative single-cell T-cell genomes illustrating T-Cell Receptor alpha (TCRa) and beta (TCRb) deletions. (**B**) Genome-wide copy number view of representative single-cell B-cell genomes illustrating light (IGL) and heavy (IGH) immunoglobulin deletions. (**C**) Bar plot quantification of T-cells, B-cells, pseudo-diploid cells and other non-tumor cells identified in profiled tumor biopsies. (**D**) Zoomed in views of TCRa deletion breakpoints in single T-cells found in tumors biopsies (P15 and P9), blood purified CD8+/4+ T-cells, and single-leukemic cells derived from a T-cell leukemia. (**E**) Representative genome-wide copy number plot of a T-cell exhibiting X-chromosome loss. Insert – bar plot quantification of X-loss T-cells in ER+ and ER- tumors. Asterisk denotes statistical significance based on chi-square test (p-value=0.0047). (**F**) Left panel: Frequency plot of CNAs identified in a panel of 200 breast cancer genomes. Highly recurrent alterations, such as 1q gain and 16 p loss are noted. Right Panel: Representative copy number plots of pseudo-diploid single-cell genomes illustrating the occurrence of recurrent breast cancer CNAs in these cells.

The online version of this article includes the following source data and figure supplement(s) for figure 1:

**Source data 1.** Patient sample associated metadata.
**Figure supplement 1.** Characteristics of tumor samples profiled in the study.
**Figure supplement 2.** Single-cell sequencing analysis of T-cells retrieved from purified lymphocytes and T-cells identified in breast cancer tissue biopsies.
**Figure supplement 3.** Copy number alterations in pseudo-diploid cells identified in tumor samples.

infiltration limited to a subset of tumors, (*Figure 1C*) consistent with previous studies (*Ruffell et al., 2012*).

Given the importance of T-cells in emerging immunotherapies (*Sharma and Allison, 2015*), we performed more detailed analysis on the identified T-cells and queried whether they represent clonally expanded T-cells that have infiltrated the tumor mass, or TCRa unique, non-clonal T-cells. To gauge the specificity in detecting genetically unique T-cells using single-cell sequencing we examined the pattern of breakpoints resulting from TCRa deletions. In the purified CD4+/CD8+ T-cell nuclei (n = 95), deletion breakpoints were found distributed over 14 genomic bins and in the majority of T-cells were not shared, suggesting unique TCRa recombination events (*Figure 1D* and *Figure 1—figure supplement 2D*). Extending this breakpoint analysis to T-cells identified in patient biopsies, we found that within a given tumor none of the T-cells shared identical breakpoints, similarly suggesting that the identified T-cells were non-clonally derived and genetically distinct. This is in contrast to the breakpoints observed in nuclei derived from a T-cell leukemia which were recurrent and positionally identical across different single nuclei (*Figure 1D* and *Figure 1—figure supplement 2E*). To quantitatively corroborate these observations, we devised a breakpoint distance metric and applied it to the abovementioned T-cell groupings (Methods). Indeed, we find lower breakpoint

pairwise-distance values in the single-nuclei derived from the T-cell leukemia compared to both CD8 +/CD4+ T-cells and T-cells found in breast cancer biopsies (*Figure 1—figure supplement 2F*). All the T-cell leukemia nuclei in this analysis shared identical breakpoints, p-value<$10^{-6}$, in contrast to T-cells found in tumor biopsies. In addition to the heterogeneity of the TCRa deletion breakpoints in T-cells, we observed large karyotypic abnormalities affecting some T-cell genomes (*Figure 1—figure supplement 2G*). These karyotypic abnormalities were represented overwhelmingly by a loss of an entire copy of the X chromosome (*Figure 1E*) consistent with previous karyotyping studies of proliferating lymphocytes (*Nowinski et al., 1990*). These X-loss T-cells, when found in the same patient sample, were genetically distinct at their respective TCRa locus as judged by the deletion breakpoints as well as the above mentioned breakpoint-distance metric (*Figure 1—figure supplement 2E and F*). Interestingly, T-cells displaying X-loss were found more frequently in estrogen receptor negative (ER-) compared to estrogen receptor positive (ER+) tumors (p-value=0.0047) (*Figure 1E*, insert). Given that BRCA1 has been shown experimentally to be associated with the process of X-inactivation (*Ganesan et al., 2002*) and that BRCA1-mutant tumors are mostly ER- tumors, this association is intriguing.

Additionally, we and others have reported the detection of individual, aneuploid nuclei that carry non-clonal, genetically heterogeneous CNAs termed pseudo-diploids (*Navin et al., 2011*; *Demeulemeester et al., 2016*). Initially identified in two triple negative primary tumors and not their matching metastasis (*Navin et al., 2011*), here, we extend their observation to tumors from all four PAM50 subtypes studied (i.e. LumA, LumB, HER2+, and Basal) (*Figure 1c*). Interestingly, we find that some carry prototypical breast cancer CNAs such as gain of 1q and loss of 16 p (*Figure 1F* – right panel). The pseudo-diploids observed in our cohort however do not appear to be precursors to the clonal cancer cells constituting the tumor mass as they do not share their respective CNAs (*Figure 1—figure supplement 3*). We interpret the observation of these nuclei as a manifestation of the inherent genomic instability present in normal breast epithelial cells of cancer patients. Further analyses, across a larger number of single pseudo-diploid nuclei while also factoring somatic single-nucleotide variant (SNV) information, are required to definitively prove this.

Thus, using single-cell sequencing we were able to detect immune cells present within the tumor mass and infer their genetic heterogeneity as well as extend the identification of pseudo-diploids to all PAM50 breast cancer subtypes and observe the occurrence of cancer associated CNAs in these cells.

## Genetic heterogeneity of large copy number alterations in cancer cells and the importance of dosage

We then proceeded to analyze the genetic heterogeneity of large-scale CNAs present in sequenced single cancer nuclei. These events include CNAs larger than 3 MB as well as whole chromosome or arm level events. As a way of illustrating the data, we plot all single-nuclei copy number profiles from a given patient on the same graph. Doing so for tumor P5 (LumA), one can visually distinguish clonal from sub-clonal alterations by observing either one or multiple copy number states at any given genomic position, respectively (*Figure 2A*). For example, losses of 8 p and 18 p (recurrent CNAs in breast cancers) are found at one copy number state and are thus clonal events (*Figure 2A* – red arrows). Alternatively, gains at 1q and 8q (also recurrent breast cancer CNAs) as well as losses on chromosomes 13 and X are found to be sub-clonal (*Figure 2A* – blue arrows). For P5, computing the fraction of the genome found to be clonal/sub-clonal using a stringent threshold (minimum 10% of cells showing sub-clonality – Materials and methods), we find P5 to be sub-clonal in over half of its genome (56% sub-clonal) (*Figure 2A* - top bar). Extending this analysis to all tumors, we find a wide range of fractional sub-clonal values with tumors showing: (1) significant (sub-clonal at ~>40% genome – Ex: P5, P22, and P59), (2) moderate (5%–20% subclonal– Ex: P14, P58, and P31) or (3) low levels of heterogeneity (subclonal at <5% of genome – Ex: P16, P41, P1) (*Figure 2B*). Representative examples of significant, moderate, and low level CNA heterogeneity samples, as illustrated for P5, are provided (*Figure 2—figure supplement 1*).

Importantly, regions found to be sub-clonal in tumor samples included regions: (1) known to be recurrently altered by CNAs (ex: 8 p and 13q), (2) harboring genes encoding therapeutic targets such as ESR1, CYP19A (*Ellis et al., 2012*), CDK6 (*Turner et al., 2015*), and PD-L1 (*Nanda et al., 2016*), (3) harboring genes known to be recurrently mutated by single nucleotide variants (SNVs) such as PIK3R1, PDGFRA, and RUNX1 (*Cancer Genome Atlas Network, 2012*; *Nik-Zainal et al.,*

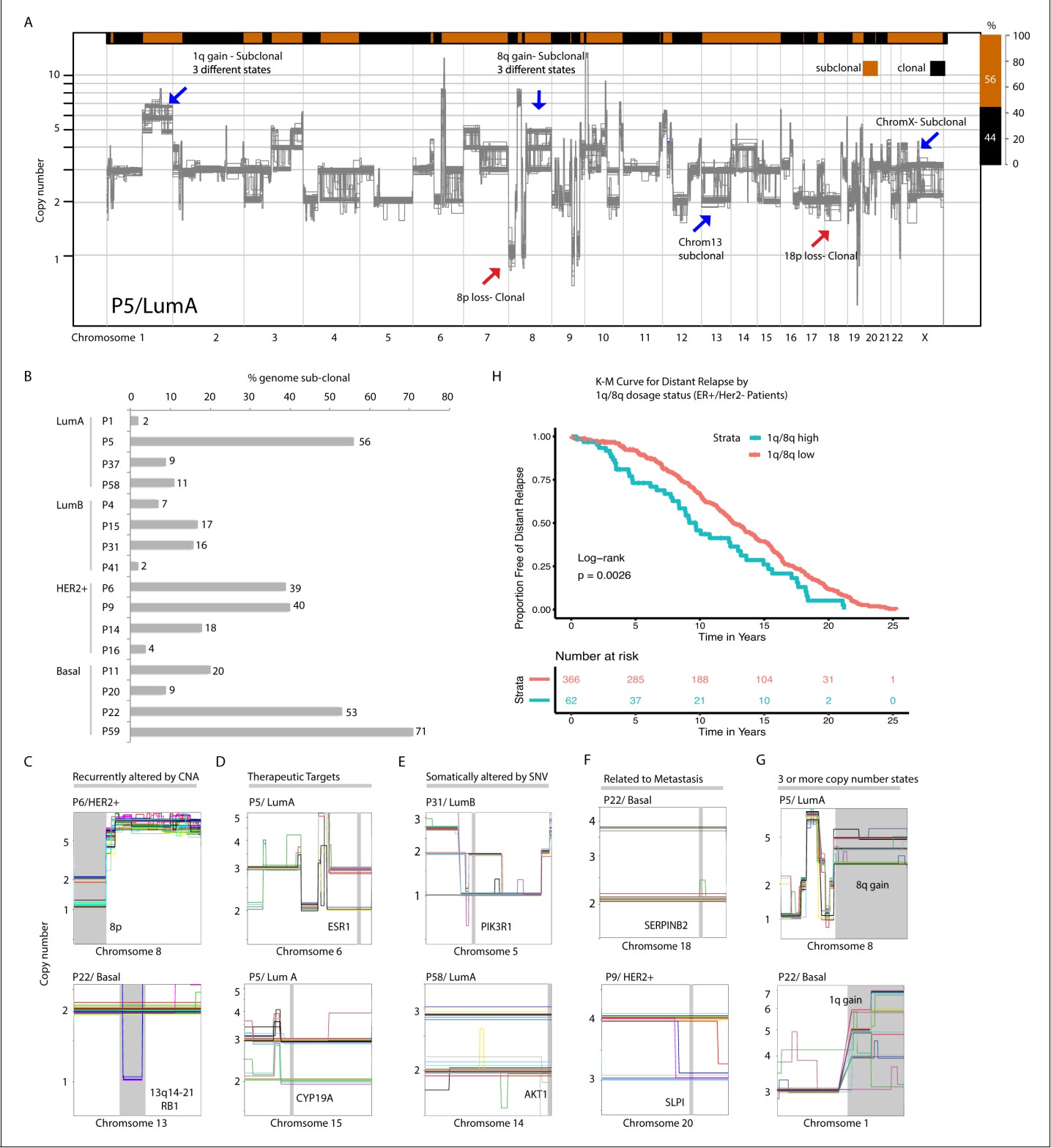

**Figure 2.** Copy number heterogeneity impacts a significant proportion of breast cancer genomes and occurs at regions in the genome that are biologically and clinically important. (**A**) Schematic illustration of CNA heterogeneity in tumor P5. Single-cell genomes (n = 131) are plotted on the same copy number diagram. Heterogeneous/sub-clonal regions can be identified by the presence of multiple copy number states (ex: 1q and chromosome 13). Red arrows point to representative clonal alterations. Blue arrows point to representative sub-clonal alterations. (**B**) Quantification of copy number heterogeneity as fraction (%) of the genome found to be sub-clonal across all sequenced biopsies. (**C–H**) Representative chromosome wide views of

*Figure 2 continued on next page*

*Figure 2 continued*

identified sub-clonal CNAs affecting: regions recurrently altered in breast cancer genomes by CNAs (C), regions containing genes of therapeutic relevance (D), regions containing genes known to be affected by somatic SNVs (E), regions with experimental evidence of involvement in breast cancer metastasis (F), regions found at three or more different copy number states (G). Gray vertical bars denote the location of genes or CNAs. (H) Distant relapse-free survival curves for cases with 1q and 8q gains that are ER+ and HER2- (n = 428). Cases are stratified based on their level of 1q or 8q gain (low vs. high).

The online version of this article includes the following source data and figure supplement(s) for figure 2:

**Source data 1.** Inferred integer copy number values of all single cancer nuclei sequenced for all patients.
**Figure supplement 1.** Copy number heterogeneity variably affects the genomes of breast cancers.
**Figure supplement 2.** Copy number heterogeneity occurs in regions of the genome that are important and harbor biologically relevant breast cancer genes.

*2016*), and (4) harboring genes experimentally shown to be involved in metastasis (*Valiente et al., 2014*; *Wagenblast et al., 2015*; *Ross et al., 2015*; *Figure 2C* - F and *Figure 2—figure supplement 2A*). DNA-FISH was used to validate a selected subset of these alterations (*Figure 2—figure supplement 2B*). Importantly, for some of the alterations, we find that the sub-clonal CNAs exist at three or more distinct copy number states in different single cells, for example, 8q gains in P5 (LumA) and P22 (Basal) (*Figure 2G*). This may have an effect on the level of expression/dosage of genes encoded on those chromosomes and thus may affect phenotypic heterogeneity. Reasoning that the increase of dosage of genes at 8q and/or 1q might be associated with advanced disease and hence bad prognosis, we devised an analysis approach to measure the relative dosage of these events in bulk copy number datasets and test their association with patient survival in a large, carefully annotated breast cancer dataset; METABRIC (20) (Methods). Indeed, we find that 1q/8q high tumors in ER+/HER2- patients (regardless of ploidy status of the tumor genome) are associated with worst distant relapse free survival (*Figure 2H* and *Figure 2—figure supplement 2C*). Further, among the newly discovered breast cancer subgroups (IntClust, IC) groups, we find that 1q/8q high tumors are enriched in the IC9 subgroup which is associated with high-risk of late distant relapse (*Rueda et al., 2019*; *Figure 2—figure supplement 2D*).

Together, these data show that heterogeneity of large copy number events can: (1) affect a large proportion of the genome in any given tumor, (2) exist at multiple levels (ex: dosage - three or more copy number states in the same tumor), and (3) that this heterogeneity can affect regions of the genome that are important for treatment or disease relapse.

## Multiple forms of genetic heterogeneity of focal chromosomal amplicons

Focal amplifications and deletions comprise another important class of CNAs. Prototypical driver amplifications containing ERBB2, CCND1, MYC, and CCNE1, as well as less commonly identified amplifications such as PPM1D and MDM2, were identified in our patient cohort and in some cases were clonal (i.e. identified in all single-nuclei sequenced) (*Figure 3A* and *Figure 3—figure supplement 1A*). However, as seen with larger CNAs, many tumors displayed chromosome amplifications genetic heterogeneity. For example in tumor P22 (basal), three focal amplicons encompassing the VEGFA, MYC and CCNE1 loci were found in varying proportions and in certain instances (VEGFA and CCNE1) in a mutually exclusive manner in sequenced single-nuclei (p-value=0.003 – Fisher's Exact Test). A sub-population lacking any of the amplicons, but having lost an additional copy of the RB1 gene via a focal deletion was also identified (*Figure 3B and C* and *Figure 3* – figure supplement B-D). Interestingly, the amplifications segregate with genetically distinct, geographically resolved tumor sub-populations (*Figure 3—figure supplement 1D and E*). Thus, P22 amplifications are somatically mosaic. Other examples of amplicon mosaicism include P5 (LumA) where an amplification targeting GATA6 was found only in a sub-population of cancer cells and P59 (Basal), where amplifications on multiple chromosomes (chromosomes 3, 5, 6 and 18) targeting important genes such as LOX (*Cox et al., 2015*) and PRDM1 (*Nik-Zainal et al., 2016*) where found in the majority of cells but absent in others (*Figure 3—figure supplement 2A*). In total, 6 out of the 16 (37%) tumors analyzed displayed amplicon mosaicism in the form of presence/absence of one or more amplicons in different subpopulations of cancer cells.

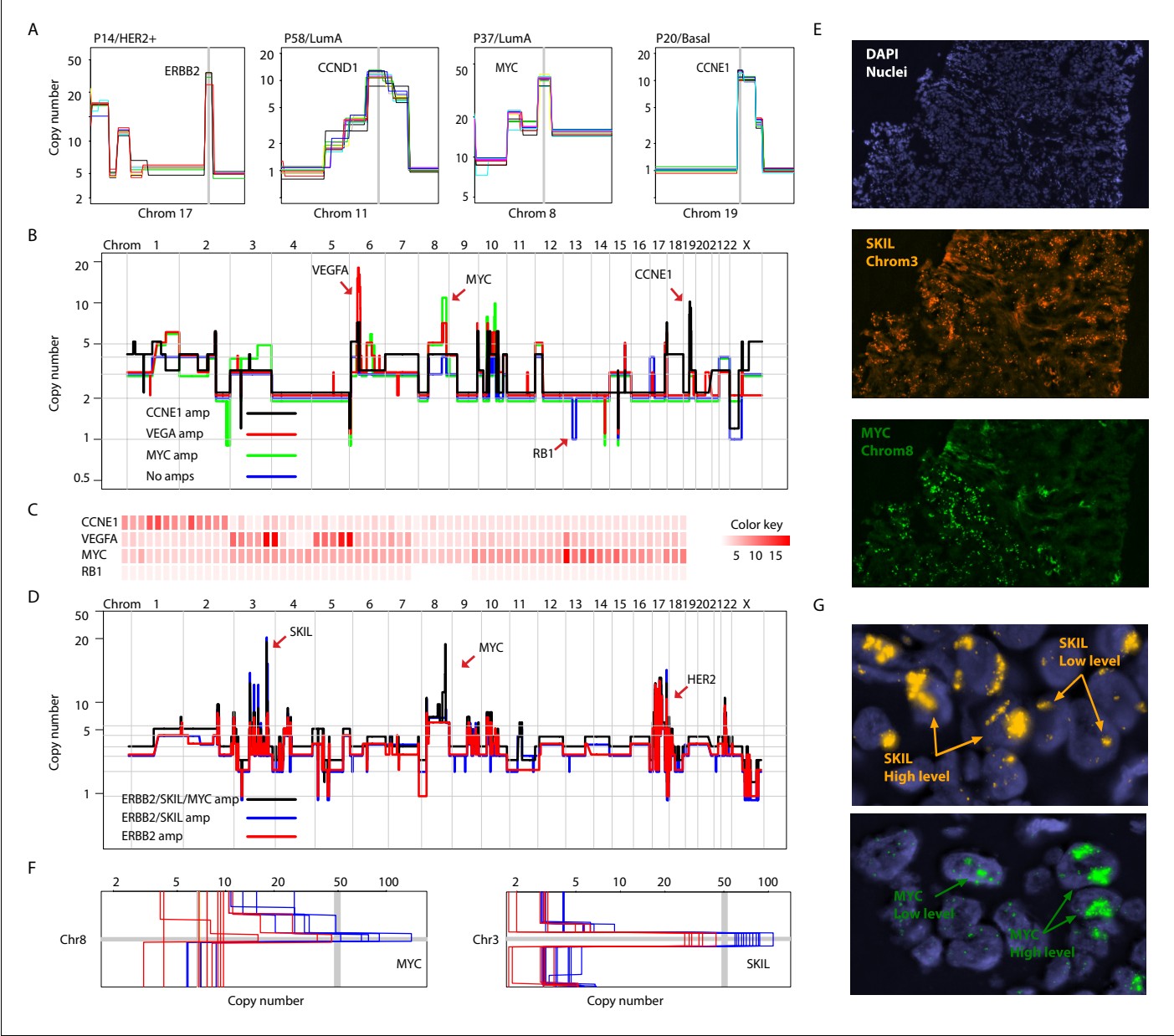

**Figure 3.** Different forms of genetic heterogeneity in chromosomal amplifications. (**A**) Representative zoom-in chromosome views of clonal prototypical breast cancer amplicons such as ERBB2 and CCND1 identified in patient samples. (**B**) Genome-wide copy number profiles of representative (n = 4) single-cell genomes illustrating heterogeneous amplifications in tumor P22. Profiles are annotated with distinguishing amplicons. A representative profile of a single nucleus with no amplifications, but deleted for the RB1 gene, is also noted. (**C**) Heatmap illustration of P22 amplicon copy number heterogeneity and the mutually exclusive nature of the CCNE1 and VEGFA amplicons - statistical significance based on Fisher's Exact Test (p-value=0.0032, Odds ratio = 0, 95% confidence interval 0–0.44). Single-nuclei are ordered according to hierarchal clustering as illustrated in *Figure 3—figure supplement 1D*. Color key is according to increasing copy number states. (**D**) Genome-wide copy number profiles of representative single-cell genomes illustrating heterogeneous amplifications in tumor P6. Profiles with distinguishing amplifications are annotated on the figure. (**E**) DNA-FISH validation of somatic mosaicism of the SKIL and MYC amplicons in tumor P6. DNA-FISH data also illustrate the geographic demarcation of the amplification in the field of view. (**F**) Zoom-in-views of the SKIL and MYC amplicons in representative single-cells illustrating the heterogeneity in the level of chromosomal amplification in tumor P6. (**G**) DNA-FISH validation of low level/high-level amplification of SKIL and MYC loci inferred based on fluorescence signal intensity.

The online version of this article includes the following figure supplement(s) for figure 3:

**Figure supplement 1.** Clonal and sub-clonal Chromosomal amplicons are detectable using single-cell copy number profiling.

**Figure supplement 2.** Examples of somatic mosaicism and differential dosage of chromosomal amplifications identified in analyzed breast cancers biopsies and identification of clonal homozygous deletions in analyzed samples.

Presence or absence of an amplicon in single cells was not the only form of variation we observed in chromosomal amplifications. Another form of variation observed came in the form of the level (or dosage) of amplification. An example is P6 (HER2+) where different spatially resolved sub-populations were identified carrying either: (1) only ERBB2 amplification, (2) ERBB2 and SKIL amplifications, or (3) ERBB2, SKIL, and MYC amplifications (*Figure 3D and E*). Importantly, for the SKIL and MYC amplicons, different single-nuclei were found to have varying levels/dosage of amplification. Where in some single-nuclei, SKIL and MYC levels reached over 60 copies, in others the level of amplification was less pronounced. For the MYC locus for example certain subpopulations contained MYC amplifications at less than 30 copies (*Figure 3F*). This was confirmed using DNA-FISH analysis based on fluorescence signal intensity (*Figure 3G*). This heterogeneity in amplicon level/dosage was also observed for the ERBB2 locus in another one of the six HER2 amplified tumors we analyzed (*Figure 3—figure supplement 2B and C*). Thus, chromosomal amplicons can exist at different levels within different single cells in a tumor mass.

We also observed homozygous deletions affecting known breast cancer genes such as MLLT4, MAP2K2, and NCoR1, among others (*Cancer Genome Atlas Network, 2012*; *Nik-Zainal et al., 2016*; *Figure 3—figure supplement 2D*). However, we did not observe heterogeneity in this class of CNAs. We cannot rule out genetic heterogeneity of this form of variation since homozygous deletions are generally smaller in size than chromosomal amplifications and at the resolution of our analysis may have gone undetected. Nonetheless, our results illustrate varied forms of genetic heterogeneity in chromosomal amplicons affecting important breast cancer genes in a significant proportion of breast tumors. These observations are particularly important given that driver cancer genes commonly found in amplicons are generally perceived to be good targets for drug development (*Cancer Genome Atlas Network, 2012*), have been found to be associated with metastatic breast cancer (*Bertucci et al., 2019*), and because sub-clonality of amplifications is difficult to infer from bulk sequence data (especially targeted sequencing) given its quantitative nature, as opposed to qualitative nature of SNVs.

## Single-cell genome sequencing and multi-region sampling yield complementary information

Previous studies utilizing multi-region sequencing to investigate somatic SNVs in breast and lung cancer have shown that a significant proportion of the variation that is found between different spatially resolved biopsies can be detected at the sub-clonal level in one of the biopsies with deeper sequencing (*de Bruin et al., 2014*; *Zhang et al., 2014*; *Yates et al., 2015*). This has not been investigated in a genome-wide, unbiased manner for CNAs. For 9 of the 16 tumors analyzed, we were able to sequence and compare bulk DNA from two spatially resolved biopsies for CNAs. In concordance with previous studies, we find substantial differences in CNAs between biopsies (*Yates et al., 2015*). Interestingly however, we find that much of the variation observed between the two biopsies can also be observed as sub-clonal variation in the single-cell data (*Figure 4A* and *Figure 4—figure supplement 1*). For example, in tumor P22 (Basal), differences on chromosomes 8, 12, 15 and X were observed between the two bulk profiles but were also observed sub-clonally at the single-cell level (*Figure 4A* and *Figure 4—figure supplement 1A and B*). Of 54 alterations differentially found in the two analyzed biopsies from the nine patients, 35 (65%) were identified in the data from single nuclei. This was observed for broad copy number events as well as focal amplifications, for example CCNE1 and GATA3 (*Figure 4B*). Importantly, additional heterogeneous CNAs were detected only in the single-cell data and not in the bulk comparisons, for example variation on chromosome 1q and 1p in P22 as well as alterations on 1q and chromosome four in P5, with the converse also being true (*Figure 4A* and *Figure 4—figure supplement 1*). Thus, in our cohort, a substantial proportion (but not all) of variation in CNAs between spatially resolved biopsies is detected at the single-cell level with higher resolution analysis and some variation is only observed at the single-cell level or via multi-region sequencing.

## CNA heterogeneity is associated with genetic, molecular, and clinical classifications

To associate CNA heterogeneity with molecular and clinical parameters we utilized two metrics from the single-cell data, applying stringent thresholds (Methods). The first metric is the fraction of the

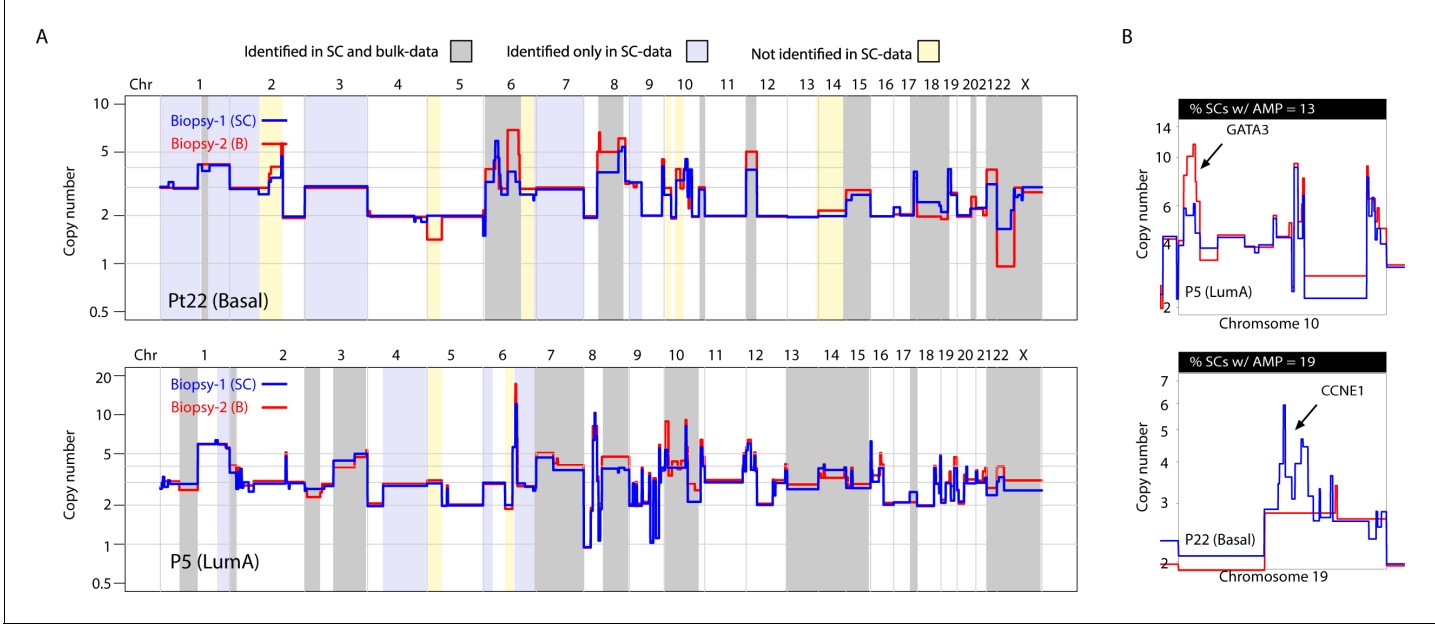

**Figure 4.** A substantial amount of copy number variation observed between spatially resolved biopsies identified in bulk sequencing data is captured at the single-cell level in one of the biopsies. (**A**) Genome-wide copy number view of bulk copy number profiles from two different geographic biopsies (blue – biopsy analyzed at bulk and single cell level, red – biopsy analyzed only in bulk) from tumor P5 (top panel) and tumor P22 (bottom panel). Gray shading denotes variation identified at bulk level between the spatially resolved biopsies and captured in single-cell data. Yellow shading indicates variation not seen in single-cell data but found in the bulk comparison. Lavender shading indicates variation observed only at the single-cell level. (**B**) Chromosomal view of representative examples of spatial variation of chromosomal amplicons found heterogeneously at bulk and single-cell level. Top panel indicates % of single-cells with the respective chromosomal amplification in single-cell data.

The online version of this article includes the following figure supplement(s) for figure 4:

**Figure supplement 1.** CNA heterogeneity identified from the analysis of two spatially resolved tumor biopsies from the same patient is commonly identified in the single-cell copy number data from one of the biopsies.

genome found to be sub-clonal. The second metric is the number of recurrent, sub-clonal break-points. We chose four discrete variables for association analysis: PAM50 subtype annotation and ploidy status (biological parameters) and HER2 and estrogen receptor status (clinical parameters). Plotting the data in bar graph format with tumors rank ordered according to increasing levels of heterogeneity while applying Wilcoxon rank order testing for associations (*Figure 5A* and *Figure 5—figure supplement 1A–C*) reveals several points. (1) Tumor samples of different PAM50 cancer subtypes display variable levels of CNA heterogeneity (i.e: inter-tumor CNA heterogeneity). For example, both P9 and P16 belong to the HER2+ subtype but differ markedly in their CNA heterogeneity profiles. Similarly, both P5 and P1 belong to the LumA subtype but display notable differences in CNA heterogeneity. Thus, for any given subtype, there appears to be a range in terms of CNA heterogeneity in the analyzed samples. We cannot associate or rule out an association between heterogeneity and any particular subtype due to the size of our cohort. Increasing sample sizes as well as analyzing multiple biopsies per sample in future studies will be required to delineate this. (2) Polyploid tumors are significantly more likely to be heterogeneous on the copy number level and this relationship is unrelated to the total number of clonal alterations found in each tumor. (3) As a group ER- tumors are significantly more heterogeneous on the copy number level than ER+ tumors. Thus copy number heterogeneity is variable within breast cancer tumors of the same subtype and is associated with tumor polyploidy and ER- disease. There is one exceptional outlier, an ER+/LumA (P5) with extensive copy number and clonal heterogeneity. Interestingly, unlike most ER+ tumors, P5 is mutated for the TP53 gene (*Figure 5B*), as are the majority of ER- tumors.

We also attempted to associate copy number heterogeneity across all samples with continuous variables such as patient age and tumor size and found no significant relationships (*Figure 5—figure supplement 1D*). However, when restricting the analysis to ER+ tumors (and excluding the ER+/TP53 mutated case) we find a relationship where tumor size and CNA heterogeneity are inversely

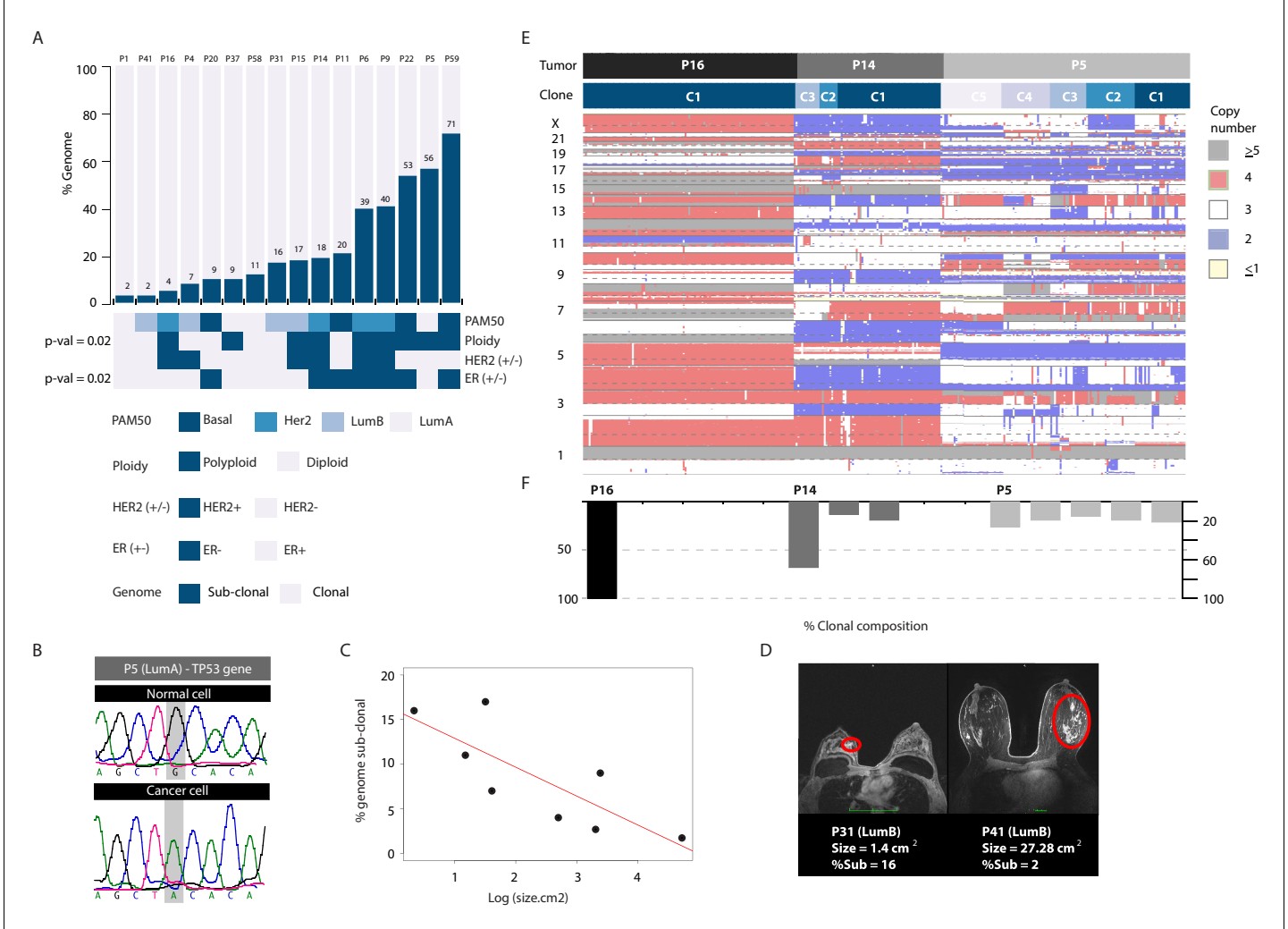

**Figure 5.** Copy number heterogeneity is associated with biological and clinical parameters and tumors can be classified according to clonal composition. (**A**) Bar plot with rank order illustrating CNA heterogeneity as measured by fraction (%) of the genome found to be sub-clonal. Each tumor is denoted at the top of each bar with values of fraction (%) genome sub-clonal also annotated. Categories for association analyses are depicted in panel below. Asterisks denote categories with statistically significant associations based on Wilcoxon rank order tests (p-value=0.0229 and 0.0164 for ER- and polyploidy categories respectively). p-values were rounded and annotated in the figure panel. (**B**) A representative illustration of a Sanger sequencing trace of DNA from a single normal and single cancer cell from tumor P5, an ER+ tumor. Traces illustrate a G to A transition at position 7578500 of chromosome 17 resulting in a non-sense mutation in the TP53 gene. (**C**) Correlation analysis between fraction (%) of genome sub-clonal and the log of the tumor size (cm2) as measured by MRI. Pearson's rank correlation p-value=0.01912. Log of the size was used given the assumption of exponential tumor growth. P5, an ER+, TP53 mutant outlier is excluded from the analysis. (**D**) Representative MRI images of two patient tumors analyzed in this study illustrating the inverse correlation of tumor size and fraction of genome sub-clonal in ER+ tumors. Red colored ellipses denote tumor positions. (**E**) Heatmap illustration of three representative tumors displaying different clonal composition patterns. Patterns are: homogenous (P16), heterogeneous with a dominant clone (P14), and heterogeneous without a dominant clone (P5). Top bar denotes tumor identity with bottom bar depicting sub-clones. Heatmap color scheme is provided to the right of the heatmap. (**F**) Quantification and bar plot illustration of clonal composition as explained in E.

The online version of this article includes the following figure supplement(s) for figure 5:

**Figure supplement 1.** Copy number heterogeneity is associated with polyploidy and estrogen receptor negative status and these associations are not directly related to the amount of rearrangements found in the cancer genomes of analyzed samples.

**Figure supplement 2.** Clonal composition is variable in breast cancer biopsies and can be used in the classification of tumors.

correlated (p-value=0.02793). The largest four tumors (measuring at more than 10 cm2) display the least amount of CNA heterogeneity (*Figure 5C and D*). This could be explained by the expansion and clonal sweep of a dominant clone in these tumors that is then captured when biopsying a geographically restricted region of a large tumor mass.

Lastly, we examined the clonal composition of cancer cells from analyzed tumor samples (i.e. # of genetically distinct clones in a given sample and their relative frequencies) and find that tumor samples in general conform to three different clonal classes: (1) homogenous tumors where only a single clone is observed, found only in ER+ tumors; (2) heterogeneous tumors where multiple clones exist but with one clone dominant over others (i.e. dominant >50%), a pattern found throughout the cohort subtypes; and (3) heterogeneous tumors where many clones are found at varying frequencies with none of them being dominant, mainly in ER- tumors (*Figure 5E and F* and *Figure 5—figure supplement 2*).

## Discussion

We provide a comprehensive analysis of breast cancer copy number alterations at the single cell level, and in the process, make many novel observations regarding the genetic heterogeneity of breast tumors. First, we observe an association between ER- tumors and T-cells exhibiting X-chromosome loss. This observation is at present of unclear clinical significance, but nevertheless might be a useful marker for this subtype of disease. Second, we observe pseudo-diploid single cells in the majority of our studied tumors, across different PAM50 subtypes, providing further evidence for the existence of such cells in breast cancer tissue. This, coupled with the observation of cancer-specific alterations (e.g. 1q gain and 16q loss) in these cells raises the question of whether these cells are a manifestation of the intrinsic genomic instability of the mammary epithelial cells of breast cancer patients and whether their detection might be useful in early stage risk assessment. Third, we observe profound genetic heterogeneity in CNAs affecting genes and regions with known biological relevance in breast cancer, such as genes associated with metastasis and therapeutic response. Thus, many cancer phenotypes might result from the selection for sub-clonal CNAs. Fourth, we find that CNA heterogeneity is not binary (i.e. sub-clonal or clonal). CNAs can be found at different copy numbers in different populations of cells and that this, in the case of 1q/8q gains is associated with clinical outcome, likely a consequence of dosage dependent biology. Fifth, the observation of mosaicism in recurrent driver amplicons in a significant proportion of tumors has clinical implications given that driver amplicon genes are often targets for drug development (e.g., ERBB2, MET, and EGFR) and have been proposed to be good targets for drug development in breast cancer (*Cancer Genome Atlas Network, 2012*). Further, we find additional layers of heterogeneity in this class of CNAs: the presence or absence of an amplicon and variation in the level/dosage of amplification, which might be the result of extra-chromosomal amplicon instability (*Turner et al., 2017*). This is of importance given that recent studies have provided evidence for a role of variation in gene dosage in therapeutic resistance to targeted therapy (*Xue et al., 2017*). Sixth, the single-cell data show that a significant proportion, but not all, of the heterogeneity in CNAs found differentially between two spatially resolved biopsy samples is also captured at the single-cell level in one of the biopsies. Importantly, some variation is captured only in the single-cell data and not in the bulk comparisons. This provides a rationale for performing both spatial sampling of tumors and deep analysis of biopsy material for precision medicine applications (*Yates et al., 2015*). Seventh, we see somewhat of an inverse correlation between tumor size and CNA heterogeneity in ER+ tumors, many of which present as a single large clone. Thus, for such large ER+ tumors, extensive spatial multi-region sampling might be necessary to capture a true picture of a tumor's genetic heterogeneity. Eight, the association of CNA heterogeneity with parameters such as polyploidy and ER- status provides an indirect association of heterogeneity with clinical outcome since both parameters have been shown to associate with a worse prognosis (*Carey et al., 2006*; *Pinto et al., 2017*). Ninth, we observe that some tumors exhibit a multitude of sub-clones with no single sub-clone being dominant (largely observed in ER- tumors). This raises questions regarding the importance of clonal cooperation in the growth and evolution of breast tumors, an area for which experimental evidence has recently emerged (*Marusyk et al., 2014*). All of the above mentioned findings provide novel observations upon which more focused investigations can be based (ex: X-loss in ER- tumors and pseudo-diploid

cells) as well as a solid foundation for future, more expansive studies of CNA heterogeneity in breast, as well as other cancers.

Importantly, in contrast to bulk sequencing where elegant studies have used deep sequencing information to enable phasing and inference of genetic heterogeneity and clonality (*Shah et al., 2012*; *Nik-Zainal et al., 2012*), our study highlights the power inherent in single-cell genomic investigations for the direct observation and quantification of genetic heterogeneity. Most of the novel observations revealed by our study would not have been possible were it not for the single-cell nature of the data. For example, the identification of T-cells with X-chromosome losses and their association with ER- disease, as well as the detection and quantification of pseudo-diploid cells would have been obscured by bulk genomic analysis. Similarly the observation that CNAs can be found at more than two distinct copy number states, which we show is associated with clinical outcome, and the identification of somatic amplicon mosaicism in a substantial proportion of breast cancer biopsies is the result of the single-cell resolution of the data and importantly, has been missed by many prior bulk genomic studies. Thus, the results provide a strong rationale for the continued investment in the development of single-cell genomic technologies (which have lagged behind single-cell transcriptomic technologies) and their utilization in the study of tumor genomes (*Baslan and Hicks, 2017*) to complement bulk sequencing efforts. This will require developing approaches that help in: scaling single-cell genome library construction as well as reducing costs (for example via the implementation of microfluidics) (*Laks et al., 2019*; *Li et al., 2020*), pairing single cell genome with single cell transcriptome information (*Dey et al., 2015*; *Macaulay et al., 2015*), and working with formalin fixed paraffin embedded specimen (*Jin et al., 2015*; *Martelotto et al., 2017*), challenges which many groups have taken efforts to address. Ultimately, single-cell genomic investigations will play an important role in advancing our knowledge of breast cancer genetics and heterogeneity and in the process advance our knowledge of the genetics and biology of the disease and help in its clinical management.

## Materials and methods

### Patient cohort and tissue samples

Tissue samples were collected from 16 patients enrolled on two, neoadjuvant - phase II, open-label clinical trails conducted by the Brown University Oncology Group (BrUOG): BrUOG 211B and BrUOG 211A. Two fresh, pre-treatment core biopsies were obtained per patient, placed into optimal cutting temperature (OCT) and frozen at −80Co for subsequent processing. Patient sample associated metadata (ex: age, ER/PR/HER2 status, among other variables) are provided in *Figure 1— source data 1*.

### RNA-Seq library generation, sequencing and analysis

RNA-sequencing libraries we generated and sequenced as previously described (*Varadan et al., 2016*). For sample subtyping, a nearest centroid classifier was implemented using log2 transformed gene expression data aligned on PAM50 list for all BrUOG samples (n = 127) where RNA-seq data was available.

### Sample processing and flow cytometry for single-nuclei capture

Nuclei isolation from frozen cores was achieved by finely mincing frozen tissue, at room temperature, in 1.0 mL of NST-DAPI buffer as previously described (*Baslan et al., 2012*). Prior to sorting, nuclei suspensions were filtered with a 5 mL Falcon round-bottom tube with a cell-trainer cap and kept on wet-ice for a minimum of 30 min. Single-cell sorting was performed using a FACS AriaIIU SORP instrument (BD Biosciences) with the ACDU option (Automated Cell Deposition Unit). DAPI signal was detected by a 355 mM UV laser (450/50 band-pass filter). For tumors with bimodal ploidy distributions (indicative of polyploidy), single-nuclei were sorted from both the diploid and polyploidy distributions for processing. For tumors with only a unimodal diploid distribution, diploid nuclei were sorted and processed. Single-nuclei were deposited into 96-well plates pre-loaded with 9 uL of cell lysis buffer per well as previously described (*Baslan et al., 2012*). All sorted plates were processed on the same day for single-cell genome amplification.

## Single-nuclei whole genome amplification and sequencing

Single-nuclei were amplified using WGA4 (Sigma-Aldrich) as per manufacturers protocol. WGA DNA was sonicated to + / - 300 bps using the Covaris instrument (duty cycle −10%, Intensity −4, cycles/burst − 200 and time 80 s). WGA sonicated DNA (processed in 96 well format, i.e. 96 nuclei per plate) was end-repaired, A-tailed and ligated to 96 custom in-house developed adaptors (*Iossifov et al., 2012*). Adaptor ligated products were purified using AMPure XP beads (Beckman Coulter). All adaptor ligated single-cell barcoded libraries (n = 96) were pooled, amplified and quantified for sequencing. All pools were sequenced on the HiSeq instrument using either SR76 or SR101 sequencing.

## Single-nuclei CNA analysis of cancer cells

Single-nuclei copy number information was inferred from sequencing data as previously described (*Baslan et al., 2012*; *Baslan et al., 2015*). In brief, sequence reads were mapped to human reference genome (reference hg19), sorted, PCR duplicated removed, and subsequently indexed. Uniquely mapped reads were counted in genomic bins using a previously developed algorithm; Varbin (*Navin et al., 2011*). Read counts were then normalized and segmentation performed using circular binary segmentation (CBS). To retrieve absolute (integer) copy number information, we utilized an algorithm that assigns ploidy and associated copy number states in single-nuclei data by utilizing a least-squares fitting algorithm used in determining a multiplier value that minimizes the variance from integer copy number states (*Baslan et al., 2015*). All inferred integer copy number values of single-nuclei for all tumors (at bin sizes of ~600 KB, dividing the genome in 5000 bins/5 k) are provided in *Figure 2—source data 1*. For analysis of fraction (%) of genome sub-clonal, we developed a pipeline where for each genomic bin we counted the number of observable copy number states found in all single-nuclei sequenced from a given patient sample. For a bin to be called sub-clonal, we required the observation of a particular copy number state in any genomic bin in at least 10% of sequenced single-nuclei. Summing all sub-clonal bins and dividing by the total number of bins is used to derive the metric: % genome sub-clonal. For this analysis, the genome was divided into five thousand bins (i.e. bin resolution of ~600 kb). Derivation of breakpoint information was performed as described previously (*Alexander et al., 2018*). In brief, bin positions at which changes in segmented copy number states occur were annotated across all sequenced single-nuclei. To account for the inherent uncertainty in breakpoint bin positioning in segmentation algorithms we utilized a window of 3 bins where a breakpoint in any of those bins was treated as equal (i.e. same breakpoint). Histograms of breakpoint distributions were then constructed and any breakpoint found in less than 90% and more than 10% of all sequenced nuclei from a given tumor sample was deemed sub-clonal. All breakpoints found in over 90% of single-nuclei data were annotated as clonal pins/breakpoints.

## Bulk DNA library generation, sequencing and analysis

Bulk DNA was purified using phenol-chloroform extraction as previously described (*Baslan et al., 2015*). Bulk DNA was sonicated, end-repaired, A-tailed, and ligated to custom in-house developed adaptors. Libraries were enriched, quantified, and pooled at equimolar concentration for sequencing on the HiSeq instrument. Bulk copy number sequence analysis was performed as described above with the exception that the multiplier, utilized in the least squares-fitting algorithm, was constrained by values derived from flow cytometry data (i.e. ploidy) to achieve absolute copy number values.

## T-cell and B-cell sample processing and analysis

Purified CD8+/CD4+ T-cells and CD19+/CD27+ memory B-cells purified cells were purchased from ALLCELLS (California, USA). Cells were thawed on ice, pelleted, and re-suspended in NST-DAPI buffer for nuclei isolation. Single-nuclei (n = 96) from both sets were sorted from the diploid peak, genome amplified, and processed for multiplex sequencing as described for the cancer biopsies above.

## T-cell breakpoint analysis

The distance between TCR deletions in two nuclei is defined to be the number of bins offset between the left breakpoints plus the number of bins offset between the right breakpoints. In order to normalized for depth and noise, nuclei with at least 1 million mapped reads were each down-

sampled to exactly 1 million mapped reads and re-segmented. All T-cell leukemia nuclei share identical breakpoint bin positions. An equal number of T-cell nuclei found in breast cancer tissue with over 1 million uniquely mapped reads included in this analysis were randomly sampled in proportional sets, 1 million times. None of these 1 million random samples had all nuclei sharing identical breakpoints indicating a p-value of $<10^{-6}$.

## DNA-FISH analysis

FISH analysis was performed on frozen section using home-brew Probe. The probe mix consisted of BAC clones containing the full length target gene and labeled with Red, Green, or Orange-dUTP as indicated in in figure legends. DNA was labeled by nick translation using fluorochrome-conjugated dUTPs from Enzo Life Sciences Inc, supplied by Abbott Molecular Inc Tissue processing, hybridization, post-hybridization washing, and fluorescence detection were performed according to standard procedures established at the MSKCC Molecular Cytogenetics Core Facility. Slides were scanned using a Zeiss Axioplan 2i epifluorescence microscope equipped with CoolCube 1 CCD camera controlled by Isis 5.5.10 imaging software (MetaSystems Group Inc, Waltham, MA). Prior to hybridization on tissue section, the probe-mix was hybridized on peripheral blood from normal healthy male to ensure locus specificity. Following hybridization, the tissue was scanned through 63X to assess signal pattern and representative regions imaged (each image was a compressed/merged stack of 12 z-section images taken at 0.5 micron intervals under the Red, Green and Orange filter respectively). Analysis was performed on captured images.

## MRI imaging

Bilateral breast MRI evaluation with a dedicated breast coil (without compression) was done on a 1.5 Tesla magnet. Images were collected for 6 min at 1 min intervals for following bolus IV gadolinium administration (0.1 mmol/kg). Images were Axial 3D SPGR fat-suppressed T1-weighted. A board certified radiologist interpreted the images.

## Bulk copy number analysis of METABRIC datasets for survival analysis

We used published copy number data from the METABRIC cohort (*Curtis et al., 2012*) and associated distant relapse free survival data to assess the association between higher dosages of 1q or 8q and patient outcomes. For 1992 cases (from the discovery and validation sets in the initial study), segmented absolute copy number calls were derived using circular binary segmentation. Specifically, the copy number data were smoothed and analyzed with the R package DNAcopy using default parameters, followed by applying the MergeLevels algorithm to the segmented data. In order to remove the dependence between cellularity and the proportion of alterations, we employed different thresholds for calling alterations and high-levels events according to the cellularity of each sample, as previously reported. The clinical outcome analysis included only cases from the original cohort that possessed a 16q loss and either a 1q gain or an 8q gain (446 cases total). Copy number status of the three chromosome arms of interest (1q, 8q, and 16q) was determined by calculating a weighted arithmetic mean of the segments comprising the chromosome arm. In order to allow for parallel analysis of diploid and polyploid tumors, the mean copy number for the 16q chromosome arm (loss of 16q is cytogenetically linked to 1q/8q gain) was used to normalize the degree of gain present in the 1q and 8q arms. For each case, the maximum of the normalized mean copy number from either 1q or 8q was used for downstream analysis. Cases were divided into those with evidence of high-level Gain on 1q or 8q (66 cases, normalized CN >1.5) and those with low-level Gain (380 cases). Kaplan-Meier analysis was performed to compare these groups within all breast cancer patients using time to distant relapse as the outcome. Since overexpression of the ER and HER2 receptors and lymph node status are strong independent predictors of clinical outcome, we also stratified our findings by ER/HER2 status and lymph node status. In a Cox proportional hazards models that accounts for other clinical covariates (age, grade, size, LN status, ER, PR, and HER2 expression) 1q/8q ratio was not independently significant.

## Statistical analysis

All statistical tests were performed using the statistical package R. For the X-loss in T-cells comparison in ER- and ER+ disease a chi-square test was used. For analysis of mutual exclusivity of CCNE1

and VEGFA amplification (copy number higher than 5) in P22, Fisher's Exact Test was used with the alterative hypothesis that true odds ratio is less than 1. For correlations of copy number heterogeneity with biological and clinical variables, Wilcoxon rank order tests were implemented. Spearman rank was used to study the association of tumor size with CNA heterogeneity in ER+ cases. p-values are provided for all statistical tests either in the figures, figure legends, or both.

## Sanger sequencing

Normal and cancer single-nuclei WGA DNA (n = 10 for each category) was subjected to PCR amplification of tp53 exons followed by Sanger sequencing. In brief, 50 ng of WGA DNA for 10 nuclei per above category were amplified in a PCR reaction using 1X Amplitaq 360 master mix with 0.2 nM of forward and reverse primers. Resultant PCR products were sequenced using standard Sanger sequencing protocols. A Sanger trace from one representative WGA DNA (one for each normal and cancer) is illustrated in *Figure 5B*.

## Data and code availability

Data generated for this study are available thought Short Read Archive (SRA) under BioProject accession number PRJNA555560. All single-cell raw sequencing data were processed using code provided in detailed in *Baslan et al. (2012)*. The R Source code for the calculation of % of genome sub-clonal is included as *Source code 1*. The R source code used for the derivation of clonal/sub-clonal pins, as described in Materials and methods section, is available on GitHub at https://github.com/jysonganan/SCclust/blob/master/R/selectpin.R.

# Acknowledgements

We thank Pamela Moody of the CSHL flow cytometry core for assistance in single-cell sorting. We thank Michael Berger, Maurizio Scaltriti, Ben Stanger, and John Morris The Fourth for critical reading of the manuscript and feedback. This work was supported by grants to MW from the Department of Defense (W81XWH-11–10747), the Breast Cancer Research Foundation (BCRF), and the McFarland Fund. MW is an American Cancer Society Research Professor. JH is supported by the BCRF and the Susan G Komen Foundation (IIR13265578). TB is supported the by the William C and Joyce C O'Neil Charitable Trust, Memorial Sloan Kettering Single Cell Sequencing Initiative. MSK Molecular Cytogenetics Core is supported by the NIH-CCSG (P30 -CA008748).

# Additional information

### Competing interests

Jie Wu, Nevenka Dimitrova: is an employee of Philips Research North America. Assaf Gordon: is affiliated with House Gordon Software Company LTD. The author has no other competing interests to declare. The other authors declare that no competing interests exist.

### Funding

| Funder | Grant reference number | Author |
| --- | --- | --- |
| Department of Defense | W81XWH-11-10747 | Michael Wigler |
| Breast Cancer Research Foundation | | James Hicks<br>Michael Wigler |
| Philips Research North America | | James Hicks |
| NIH | P30 -CA008748 | Gouri Nanjangud |
| William C and Joyce C O'Neil Charitable Trust | | Timour Baslan |
| McFarland Fund | | Michael Wigler |
| American Cancer Society | Research professor | Michael Wigler |

| Susan G. Komen | IIR13265578 | James Hicks |
| --- | --- | --- |
| Memorial Sloan-Kettering Cancer Center | Single Cell Sequencing Initiative | Timour Baslan |

The funders had no role in study design, data collection and interpretation, or the decision to submit the work for publication.

## Author contributions

Timour Baslan, Conceptualization, Resources, Data curation, Software, Formal analysis, Supervision, Validation, Investigation, Visualization, Methodology, Project administration; Jude Kendall, Data curation, Software, Formal analysis, Investigation, Visualization; Konstantin Volyanskyy, Software, Formal analysis, Investigation, Visualization; Katherine McNamara, Formal analysis, Investigation, Visualization; Hilary Cox, Sean D'Italia, Frank Ambrosio, Michael Riggs, Data curation, Methodology; Linda Rodgers, Data curation, Formal analysis, Methodology; Anthony Leotta, Data curation, Software, Formal analysis, Supervision, Visualization; Junyan Song, Software, Formal analysis; Yong Mao, Data curation, Software, Formal analysis, Visualization; Jie Wu, Data curation; Ronak Shah, Formal analysis; Rodrigo Gularte-Mérida, Validation, Investigation, Visualization, Methodology; Kalyani Chadalavada, Methodology; Gouri Nanjangud, Formal analysis, Validation, Investigation, Visualization, Methodology; Vinay Varadan, Data curation, Formal analysis; Assaf Gordon, Data curation, Visualization; Christina Curtis, Data curation, Formal analysis, Supervision; Alex Krasnitz, Resources, Data curation, Software, Formal analysis, Supervision, Visualization; Nevenka Dimitrova, Supervision, Project administration; Lyndsay Harris, Resources, Supervision, Investigation; Michael Wigler, Formal analysis, Supervision, Funding acquisition, Investigation, Methodology; James Hicks, Conceptualization, Resources, Supervision, Funding acquisition, Investigation, Methodology

## Author ORCIDs

Timour Baslan (iD) https://orcid.org/0000-0001-5674-6432
Vinay Varadan (iD) http://orcid.org/0000-0003-1520-1967

## Ethics

Human subjects: Informed consent for sample acquisition and sequencing was obtained per the BrUOG and Case Western University Institutional Review Boards by Dr. Lyndsay Harris as part of clinical trial BrUOG 211A/B. The described study was also approved by the CSHL Institutional Review Board under Project IRB-12-017 titled "Single Cell Exome & Copy Number Profiling in Cancer" lead by Dr. James Hicks.

## Decision letter and Author response

Decision letter https://doi.org/10.7554/eLife.51480.sa1
Author response https://doi.org/10.7554/eLife.51480.sa2

# Additional files

## Supplementary files

• Source code 1. R Source code for the calculation of % of genome sub-clonal.

• Transparent reporting form

## Data availability

Data generated for this study are available thought Short Read Archive (SRA) under BioProject accession number PRJNA555560.

The following dataset was generated:

| Author(s) | Year | Dataset title | Dataset URL | Database and Identifier |
|---|---|---|---|---|
| Baslan B, Kendall J, Volyanskyy K, McNamara K, Cox H, D'Italia S, Ambrosio F, Riggs M, Rodgers L, Leotta A, Song J, Mao Y, Wu J, Shah R, Gularte-Mérida R, Chadalavada K, Nanjangud G, Varadan V, Gordon A, Curtis C, Krasnitz A, Dimitrova N, Harris L, Wigler M, Hicks J | 2020 | Single-cell genome sequencing of breast cancer | https://www.ncbi.nlm.nih.gov/bioproject/PRJNA555560/ | NCBI BioProject, PRJNA555560 |

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
