## [Decision Letter]

**Acceptance summary:**

The analysis of breast cancers at the level of Copy Number Alterations to asses intra-tumoral genetic heterogeneity at the level of single-cell sequencing is an important advance, especially because the results translate into a number of interesting new biological observations, for example, X-loss in ER- tumors and pseudo-diploid cells, among others.

**Decision letter after peer review:**

Thank you for submitting your article "Novel insights into breast cancer copy number genetic heterogeneity revealed by single-cell genome sequencing" for consideration by *eLife*. Your article has been reviewed by three peer reviewers, and the evaluation has been overseen by a Reviewing Editor and Jeffrey Settleman as the Senior Editor. The following individuals involved in review of your submission have agreed to reveal their identity: Jonas Demeulemeester (Reviewer #1); Daniel Rico (Reviewer #2).

The reviewers have discussed the reviews with one another and the Reviewing Editor has drafted this decision to help you prepare a revised submission.

Summary:

Baslan et al. present an in-depth characterization of intratumor copy number heterogeneity in a cohort of 16 primary breast cancer cases: 4 each of the Luminal A, Luminal B, HER2+ and Basal subtype They also generated bulk RNA-seq and bulk genome sequencing for the same samples. The authors isolate nuclei, sort populations with distinct ploidy using FACS, and perform low-pass whole-genome sequencing on an average 116 cells per tumor. Analyses reveal intriguing patterns of both focal and whole-genome copy number heterogeneity which are likely to have clinical relevance. The previous single-cell genome sequencing study in breast cancer primary tumors only included 2 patients (Navin et al., 2011). Therefore, the new dataset presented is the most comprehensive single cell genome sequencing for breast cancer to date. However, the manuscript was not easy to read as there were many different "short stories", which may be difficult to follow for a non-specialized audience. Also, some of the analysis were not clear, which could be improved by making the code and the data available.

Essential revisions:

1) In the second paragraph of the subsection “CNAs and genetic heterogeneity of stromal, non-cancer cells”, the authors describe their breakpoint analysis. At the resolution provided by low-pass whole-genome sequencing, the inferred breakpoints are rough approximations of the true breakends at best. Indeed, authors describe using a window of 3 genomic bins in the Materials and methods. Vice versa, random noise can lead to distinct segmentation in single cells which share a breakend (e.g. the jumps from CN 1 to 2 for the leukaemic cells in Figure 1D). As such, conclusions such as "indicating unique TCRa recombination events", "none of the T-cells shared identical breakpoints" and "genetically distinct as judged by the deletion breakpoints at their respective TCRa locus" should be stated considerably more carefully.

2) Pseudo-diploid cells are identified in the samples and authors conclude they are not precursors to the main tumor clone based on their distinct CNAs. Authors should further assess whether these cells represent early, potential evolutionary dead-end tumor subclones or do indeed stem from normal breast epithelia. To do this, authors could carefully call SNVs on the pooled low-pass sequencing data and subsequently genotype the called variants in the single cells.

3) "High-throughput sequence data should be uploaded before resubmission, with a private link for reviewers provided (these are available from both GEO and ArrayExpress)" The data availability section is not very clear: is it already available, in which case I could not find the related project on SRA (= Short Read Archive, not Short Reach Achieve) nor any link to the data.

4) The analyses of the data shown on Figure 1D are not sufficiently quantitative. Breakpoints accuracy is inherently limited with shallow coverage sequencing and will correlate with technical aspects, such as number of reads, duplicates, noise in the data, etc. The authors merge breakpoints if in adjacent bins but that these four profiles on Figure 1D visually look different is not enough to persuade readers that biological rather than technical effects are driving the differences. As this is a major claim in the manuscript, the authors are encouraged to make the analyses more quantitative (at least consider the impact of one noise metric related to "depth of coverage" in the comparison, as it could be driving the differences between the batches of single nuclei).

5) Figure 2B: The more single nuclei were sequenced and included in the analyses, the more likely that some events will be considered subclonal (>=10 nuclei with a different state). Hence, before reaching any conclusions, the authors should explicitly correct the fraction of genome subclonal for this. It looks from other figures that the number of nuclei per cancer sample can vary quite substantially thus potentially influencing the subclonal fraction. As a quick check, an easy way to correct for the number of nuclei analyzed would be for each patient, to downsample to the number of nuclei in the patient with the lowest number.

6) Figure 5A same remark as Figure 2B applies.

7) It would be useful to also have the processed data available. I could not see if it is actually included in the SRA BioProject (I guess the accession ID is not yet publicly available) but it would be good to have the patient metadata, the gene expression data, CNA calls of both the bulk and the single-cell samples in supplementary data files and/or in a relevant repository for direct download. As gene expression levels and CNA calls do not contain sequence information, these data should be made available without any restrictions.

8) The results of different interesting analyses are presented to illustrate the usefulness of the data. However, I had the impression that there were several inconclusive stories: the loss of chromosome X, for example, in connection with BRCA1 and X inactivation is intriguing but it is not further investigated. The authors may want to choose if they want to further invest in making the resource data available for different types of users or choose one of the observations (there are up to nine different conclusions in the Discussion!) to extend further for a more solid and complete "Research Article".

9) Some strong claims are made regarding the association with biological subtypes and patient stratification. For example, the sub-clonality of each tumor is precisely calculated with hundreds of cells per tumor, but at the end of the day there are only 16 patients, just four per sub-group based on gene expression. Indeed, there are some statistical tests that yielded small p-values (e.g. Figures 3C and 5A) but it is unclear what where the unit of analysis – is the patient the unit of analysis or the cell? The inclusion of the R code used, together with the data, would make the interpretation of these results clearer.

[Editors' note: further revisions were suggested prior to acceptance, as described below.]

Thank you for submitting your article "Novel insights into breast cancer copy number genetic heterogeneity revealed by single-cell genome sequencing" for consideration by *eLife*. Your article has been reviewed by three peer reviewers, and the evaluation has been overseen by a Reviewing Editor and Jeffrey Settleman as the Senior Editor. The following individual involved in review of your submission has agreed to reveal their identity: Daniel Rico (Reviewer #2).

The reviewers have discussed the reviews with one another and the Reviewing Editor has drafted this decision to help you prepare a revised submission.

The revised version of the in-depth characterisation of breast cancer cases at the level of copy number heterogeneity at the cell level builds an interesting data set, represents an advanced use of the technology and provides information of potential clinical relevance.

Summary:

The revised version of the manuscript has essentially addressed all the key points (i.e. accuracy and level of resolution of the data, cell of origin and linages, and clarification of the narrative and conclusions) but the issues with data and software accessibility have only been partially addressed.

Essential revisions:

For the publication of this article, like for any other one, it is essential that the research community has the ability to the reproduce the reported results. In this case, the metadata associated with the processed data in the supplementary file is missing and the software code to reproduce the various analyses presented is not available. These are significant issues that have to be corrected.

---

## [Author Response]

Summary:Baslan et al. present an in-depth characterization of intratumor copy number heterogeneity in a cohort of 16 primary breast cancer cases: 4 each of the Luminal A, Luminal B, HER2+ and Basal subtype They also generated bulk RNA-seq and bulk genome sequencing for the same samples. The authors isolate nuclei, sort populations with distinct ploidy using FACS, and perform low-pass whole-genome sequencing on an average 116 cells per tumor. Analyses reveal intriguing patterns of both focal and whole-genome copy number heterogeneity which are likely to have clinical relevance. The previous single-cell genome sequencing study in breast cancer primary tumors only included 2 patients (Navin et al., 2011). Therefore, the new dataset presented is the most comprehensive single cell genome sequencing for breast cancer to date. However, the manuscript was not easy to read as there were many different "short stories", which may be difficult to follow for a non-specialized audience. Also, some of the analysis were not clear, which could be improved by making the code and the data available.

We thank the reviewers for their positive comments and constructive critique of our work. Below we address the requested Essential revisions.

Regarding the above comment of “the manuscript was not easy to read”: we find this perplexing as many colleagues read the work and did not communicate any sort of illegibility pertaining to the text.

Structuring the manuscript in the form of “short stories” was done purposely (and for the benefit of a non-specialized audience) because of the number and diversity of intriguing observations that we noted from the analyses.

Regardless, where we thought it might be off benefit, we have edited certain portions of the text. We hope this makes the text easier to read.

Lastly, we have addressed the unclear analyses referenced above as well as in the Essential Revisions and made all the data publicly available for anyone to download and analyze (the vast majority of the code used in the communicated manuscript is described in our previous referenced works and can be used to reproduce the data). In addition, we are providing all processed single-cell CNA calls for all single-cancer cells analyzed in this study. These are now provided as supplementary data.

Essential revisions:1) In the second paragraph of the subsection “CNAs and genetic heterogeneity of stromal, non-cancer cells”, the authors describe their breakpoint analysis. At the resolution provided by low-pass whole-genome sequencing, the inferred breakpoints are rough approximations of the true breakends at best. Indeed, authors describe using a window of 3 genomic bins in the Materials and methods. Vice versa, random noise can lead to distinct segmentation in single cells which share a breakend (e.g. the jumps from CN 1 to 2 for the leukaemic cells in Figure 1D). As such, conclusions such as "indicating unique TCRa recombination events", "none of the T-cells shared identical breakpoints" and "genetically distinct as judged by the deletion breakpoints at their respective TCRa locus" should be stated considerably more carefully.

Points (1) and (4) of Essential revisions critique the same point: TCR breakpoints and clonal lineage relationship.

Both points are thus collectively addressed here.

We agree with the reviewers that the listed statements above are factually correct:

1) “the inferred breakpoints are rough approximations of the true breakend at best”

2) “the analyses of the data shown on Figure 1D are not sufficiently quantitative”

3) “… consider the impact of one noise metric related to “depth of coverage” in the comparison…”

To address this critique, we have performed the following analyses:

For single T-cell nuclei with at least 1 million uniquely mapped reads derived from: CD8^+^/CD4^+^ purified T-cells, a clonal T-cell leukemia, and T-cells identified in breast tumor tissue (including T-cells exhibiting X-chromosome loss), we:

1) Uniformly down-sampled uniquely mapped sequencing reads to a coverage of 1 million reads.

2) Re-segmented the “sequencing-depth normalized” datasets to re-compute the breakpoint positions/boundaries for each single-nuclei.

3) We then calculated pair-wise distances of the genomic bins that lie within the deletion breakpoints for all single-nuclei of each category.

Low pair-wise distance values indicate a higher probability that the events are clonal due to underlying genetics/biology and not due to technical artifacts. The T-cell leukemia nuclei all shared identical left and right breakpoints.When performing 1 million iterations of random sampling of breakpoint positions (i.e. bins) for the same number of T-cells found in breast tumor biopsies, we find none of these random samples shared identical breakpoints indicating a p-value of < 10^-6^. This analysis is detailed in the main text, the Materials and methods section, and is illustrated in graphical form in Figure 1—figure supplement 2G.

We have also modified the language in the main text to communicate our conclusions.

It is important to stress that our T-cell breakpoint analysis was not intended to imply that all T-cells found in a tumor mass are TCRa unique and do not share a clonal lineage. This most certainly is not the case and has been reported in the literature. The purpose of the analyses was to corroborate the intriguing finding of X-loss enrichment in T-cells found in ER-negative disease compared to ER-positive disease.

2) Pseudo-diploid cells are identified in the samples and authors conclude they are not precursors to the main tumor clone based on their distinct CNAs. Authors should further assess whether these cells represent early, potential evolutionary dead-end tumor subclones or do indeed stem from normal breast epithelia. To do this, authors could carefully call SNVs on the pooled low-pass sequencing data and subsequently genotype the called variants in the single cells.

We agree that the nature of pseudo-diploid cells in interesting and warrants further investigation, particularly using a combination of SNV and CNA genotyping as pointed out by the reviewers.

Unfortunately, this is not something that is feasible with the datasets we have or the material generated in this study.

1) We did not perform Whole Exome Sequencing on our tumor biopsies to allow precise genotyping of somatic SNVs in our tumor samples (nor can we now perform the experiment for a multitude of reasons).

2) The depth at which we performed sequencing of single-nuclei (~ 2 million reads per nuclei), even when pooling clonal cancer nuclei, let alone pseudo-diploid nuclei, is not sufficient to allow confident, de-novo calling of somatic SNVs.

3) Even if we attempted to re-sequence normal, pseudo-diploid, and cancer nuclei to higher coverages, the analysis of the data will still be extremely challenging, particularly for rare events such as pseudo-diploid cells, because the WGA method we employed (DOP-PCR) does not provide coverage of the entire genome for each single nuclei WGA DNA material.

All the factors listed above, and others, unfortunately precludes us from performing the requested assessment.

However, the question posed is one of the questions an ongoing study aims to address (a study that is an extension of the work described here). In this ongoing study we have factored this specific subject matter and designed our experimental and analytical protocols to address it.

3) "High-throughput sequence data should be uploaded before resubmission, with a private link for reviewers provided (these are available from both GEO and ArrayExpress)" The data availability section is not very clear: is it already available, in which case I could not find the related project on SRA (= Short Read Archive, not Short Reach Achieve) nor any link to the data.

We sincerely apologize for this. Our intention was always to make the data publicly available prior to the review of the manuscript. The reason the data was not available (and escaped our knowledge) is because of an apparent corrupted syntax of a subset of the sequencing files that prevented “completion” of the SRA upload and consequently, data release.

All data has now been verified to be released and publicly available.

4) The analyses of the data shown on Figure 1D are not sufficiently quantitative. Breakpoints accuracy is inherently limited with shallow coverage sequencing and will correlate with technical aspects, such as number of reads, duplicates, noise in the data, etc. The authors merge breakpoints if in adjacent bins but that these four profiles on Figure 1D visually look different is not enough to persuade readers that biological rather than technical effects are driving the differences. As this is a major claim in the manuscript, the authors are encouraged to make the analyses more quantitative (at least consider the impact of one noise metric related to "depth of coverage" in the comparison, as it could be driving the differences between the batches of single nuclei).

This is addressed above. Please see point 1.

5) Figure 2B: The more single nuclei were sequenced and included in the analyses, the more likely that some events will be considered subclonal (>=10 nuclei with a different state). Hence, before reaching any conclusions, the authors should explicitly correct the fraction of genome subclonal for this. It looks from other figures that the number of nuclei per cancer sample can vary quite substantially thus potentially influencing the subclonal fraction. As a quick check, an easy way to correct for the number of nuclei analyzed would be for each patient, to downsample to the number of nuclei in the patient with the lowest number.

We thank the reviewers for bringing this issue up and naturally agree with regards to the influence of the # of single nuclei sequenced and the proportion of the genome found to be sub-clonal. We have done the requested down-sampling analysis above and find no significant differences in the computed fraction of the of the genome found to be sub-clonal for all the tumors with the exception of two (P5 and P58).

Importantly, when performing Wilcoxon rank order testing for statistical associations with discrete and continuous variables, as done in the original submitted work, the change of order of the tumors does not change the statistical significance of the results (i.e. p-values).

Thus, given that the original data (i.e. computed sub-clonal values with all single nuclei factored) represents a more accurate measurement per tumor and that the results following down-sampling normalization do not impact our conclusions, we have kept the stats (and figures) as originally reported.

6) Figure 5A same remark as Figure 2B applies.

Please see point 5 above.

7) It would be useful to also have the processed data available. I could not see if it is actually included in the SRA BioProject (I guess the accession ID is not yet publicly available) but it would be good to have the patient metadata, the gene expression data, CNA calls of both the bulk and the single-cell samples in supplementary data files and/or in a relevant repository for direct download. As gene expression levels and CNA calls do not contain sequence information, these data should be made available without any restrictions.

The critique here is similar to the critique outlined in point 3. In addition, in our revised work we are including all processed, integer copy number calls for all the cancer cells found in the tumor samples analyzed at a bin resolution of 600kb (i.e. diving the genome into 5,000 bins – 5k).

8) The results of different interesting analyses are presented to illustrate the usefulness of the data. However, I had the impression that there were several inconclusive stories: the loss of chromosome X, for example, in connection with BRCA1 and X inactivation is intriguing but it is not further investigated. The authors may want to choose if they want to further invest in making the resource data available for different types of users or choose one of the observations (there are up to nine different conclusions in the Discussion!) to extend further for a more solid and complete "Research Article".

Please see above (3 and 7) regarding data release: all data has been publicly released and is available for download. We are also including all processed data from cancer cells as supplementary data. We feel a strength of our study lies in communicating the usefulness of single-cell genomics in furthering an understanding of cancer – which is pointed out in the text. This is very important since single-cell genomics in reality has not progressed sufficiently enough in comparison to single-cell transcriptomics (due to many reasons).

Thus, one of the aims of our manuscript was to communicate the importance of single-cell genomics that would entice the research community to adopt single-cell genomics.

We feel the importance of single cell genomics was sufficiently communicated in our work by illustrating the panoply of novel genetics observations that where gleaned from the data as a whole rather than focusing on one particular observation.

Thus, it is our opinion that communicating a plethora of novel findings, and emphasizing the power of single-cell genomics in retrieving those findings, we are communicating a “complete Research Article”.

Regarding the comment about the interesting link BRCA1 and X-chromosome loss in T-cells: we agree with the reviewer. This is something we hope to further address in ongoing studies (please see point 2 above).

9) Some strong claims are made regarding the association with biological subtypes and patient stratification. For example, the sub-clonality of each tumor is precisely calculated with hundreds of cells per tumor, but at the end of the day there are only 16 patients, just four per sub-group based on gene expression. Indeed, there are some statistical tests that yielded small p-values (e.g. Figures 3C and 5A) but it is unclear what where the unit of analysis – is the patient the unit of analysis or the cell? The inclusion of the R code used, together with the data, would make the interpretation of these results clearer.

We are not entirely certain what is meant by “strong” claims. If by “strong claims” the reviewers mean “exaggeration”, we respectfully disagree.

To illustrate, regarding the association of CNA heterogeneity with molecular and clinical parameter (i.e. biological subtypes and patient stratification), in the Discussion section we explicitly state “the association of CNA heterogeneity with parameters such as polyploidy and ER- status provides *an indirect association* of heterogeneity with clinical parameters”.

“Indirect association” cannot be construed as a “strong” claim in our opinion.

Similarly, we explicitly state “we see *somewhat* of an inverse correlation between tumor size and CNA heterogeneity”. “Somewhat” does not communicate a “strong claim” either.

We are simply laying out all the observations we have been able to glean from the data in an effort to illustrate:

1) The magnitude of CNA heterogeneity in breast cancer and hypothesize about its effect on the biology of tumors.

2) The importance of single-cell genomics in studying cancer.

Regarding the p-values and the unit of the analysis comment above:

We believe it is clear that for Figure 3C, the unit is a single-cell since we are describing a single tumor.

Similarly for Figure 5A, the unit is the patient since we are discussing the tumor cohort.

[Editors' note: further revisions were suggested prior to acceptance, as described below.]

Essential revisions:For the publication of this article, like for any other one, it is essential that the research community has the ability to the reproduce the reported results. In this case, the metadata associated with the processed data in the supplementary file is missing and the software code to reproduce the various analyses presented is not available. These are significant issues that have to be corrected.

The metadata on all tumor samples analyzed is now included in the revised version (Figure 1—source data 1). The software code to process the sequencing data (i.e. derive integer copy number state) has been described in detail in the literature and is appropriately cited. The source code for deriving two metrics used in statistical associations: (1) % genome sub-clonal and (2) clonal/subclonal pins/breakpoints, are now included as a supplementary file and a URL, respectively.